# Preserving Continuity and Trust in Primary Care: Strategies for Implementing Team-Based Models in South Tyrol, Italy

**DOI:** 10.3390/ijerph22040477

**Published:** 2025-03-23

**Authors:** Christian J. Wiedermann

**Affiliations:** Institute of General Practice and Public Health, Claudiana—College of Health Professions, 39100 Bolzano, Italy; christian.wiedermann@am-mg.claudiana.bz.it

**Keywords:** continuity of care, primary care reform, general practitioners, trust, multidisciplinary care, healthcare workflows, patient engagement, South Tyrol, digital health integration

## Abstract

Continuity of care is fundamental to the efficacy of primary healthcare, fostering trust, enhancing patient satisfaction, and improving health outcomes. However, the implementation of Ministerial Decree 77/2022, which advocates for team-based care in multidisciplinary Community Health Centers, presents challenges to these established principles. This article proposes strategies to maintain continuity and trust whilst supporting the reform objectives, specifically tailored to the unique linguistic and cultural context of the Autonomous Province of Bolzano—South Tyrol. A synthesis of regional healthcare reports, academic literature, and practical insights from implementing Ministerial Decree 77/2022 was performed to develop strategies addressing challenges such as ensuring continuity, minimizing administrative burdens, and promoting patient and general practitioner engagement. Strategies include establishing Community Health Centers as integration hubs, assigning primary providers within teams, formalizing personalized care contracts, leveraging digital tools for collaboration, and expanding the roles of nurses and care coordinators. Additional measures focus on building infrastructure for telemedicine and home-based care, engaging patients through transparent communication and feedback loops, and preserving GP autonomy through flexible participation models and incentives. Strategies adapted to accommodate South Tyrol’s demographic, cultural, and systemic characteristics can maintain continuity and trust during the transition to team-based care. By addressing key risks and fostering collaboration among stakeholders, these reforms can enhance healthcare delivery without compromising the principles of personalized, patient-centered care.

## 1. Introduction

Continuity of care is a cornerstone of effective primary healthcare, fostering strong patient–clinician relationships and improving health outcomes through trust, familiarity, and consistent management of patient needs [1]. In Italy, the traditional model of independent general practitioners (GPs) serving as the primary point of contact for patients has long upheld this principle [2]. However, the introduction of Ministerial Decree 77/2022 represents a significant transformation in the country’s primary care landscape. This reform mandates the establishment of multidisciplinary Community Health Centers (Case di Comunità, CHCs) and promotes a shift toward team-based care models, aiming to address workforce shortages, enhance healthcare accessibility, and integrate digital health solutions into routine primary care delivery [3]. Ministerial Decree 77/2022, issued on 22 June 2022, reforms Italy’s National Health Service (NHS) by shifting from hospital-centered care to decentralized, community-based healthcare [4]. A key aspect is the establishment of Community Health Centers (Case di Comunità, CHCs), designed to operate 24/7 and integrate primary, specialist, and social care. The decree mandates over 1350 CHCs nationwide, staffed by multidisciplinary teams, including general practitioners, specialists, nurses, and social workers, to enhance accessibility and continuity of care. The reform also promotes digital health solutions, such as electronic health records and telemedicine, to improve care coordination, especially in underserved areas. By strengthening primary care, preventive medicine, and team-based models, the decree aims to reduce hospitalizations and foster a more resilient, patient-centered healthcare system in Italy.

While Ministerial Decree 77/2022 is a national reform, its implementation in South Tyrol—a uniquely autonomous, bilingual region in northern Italy—presents distinct challenges and opportunities. South Tyrol’s healthcare system operates within a cultural and linguistic context that differs from much of Italy, with a significant proportion of the population being German-speaking and a tradition of strong, long-term relationships between GPs and their patients [3]. The shift from independent GP-led care to CHC-based multidisciplinary teams risks disrupting these well-established relationships, introducing potential barriers related to language, bureaucracy, and patient trust [5].

The shift to interprofessional, team-based care models has been shown to enhance accessibility, patient safety, and health management [6,7,8,9]. Research indicates that collaborative care improves care coordination, reduces hospitalizations, and enhances chronic disease management. Interprofessional teams facilitate safer decision-making, reduce diagnostic errors, and improve treatment adherence. These models are particularly beneficial for aging populations and patients with complex conditions, ensuring continuity of care beyond individual providers. By integrating specialists, nurses, and allied health professionals into CHCs, they address physician shortages and improve healthcare access, particularly in rural regions like South Tyrol.

International experience shows that team-based primary care can improve access, continuity, and efficiency [8]. In Canada, Primary Care Networks and Family Health Teams have enhanced chronic disease management, reduced hospital visits, and strengthened preventive care [10]. Brazil’s Family Health Strategy has expanded primary care access and improved population health outcomes [11]. While Ministerial Decree 77/2022 presents structural challenges, its success in South Tyrol will depend on effective team integration, digital tools, and patient-centered strategies, balancing risks with proven benefits.

This paper examines strategies to maintain continuity of care amid primary care reforms in South Tyrol, emphasizing the preservation of GP–patient relationships, trust concerns, and the use of digital and collaborative tools. It offers actionable recommendations that align with reform objectives while upholding South Tyrol’s tradition of personalized healthcare. The article synthesizes evidence from regional healthcare reports, peer-reviewed literature on continuity and trust in primary care, and insights from the implementation of Ministerial Decree 77/2022. The proposed strategies are based on best practices in healthcare reform and are tailored to South Tyrol’s demographic, cultural, and systemic characteristics.

## 2. Background

### 2.1. Importance of Continuity in Primary Care

Continuity of care is a fundamental element of primary healthcare, particularly in systems centered around GPs (Figure 1). It is associated with higher levels of trust between patients and clinicians, increased patient satisfaction, better adherence to treatment plans, and reduced healthcare costs through more efficient management of chronic conditions and prevention of unnecessary hospitalizations. Strong, ongoing relationships enable GPs to gain a deeper understanding of their patients’ medical histories, social circumstances, and preferences, facilitating personalized and effective care [12,13].

However, the shift toward team-based care models, as mandated by Ministerial Decree 77/2022, introduces a potential risk of care fragmentation. While multidisciplinary approaches aim to enhance healthcare delivery by leveraging the expertise of various professionals, they may inadvertently disrupt long-standing GP–patient relationships. If patients perceive a loss of personal connection with their primary care provider, it could erode trust and compromise care outcomes [14]. In regions like South Tyrol, where GPs often serve as trusted family doctors within closely knit communities [15], maintaining continuity is particularly critical for ensuring that reforms enhance rather than diminish the quality of primary care.

### 2.2. Specific Risks in South Tyrol

The implementation of team-based care models under Ministerial Decree 77/2022 poses distinct risks for continuity of care in South Tyrol [16], rooted in the region’s unique healthcare landscape [17] and the professional culture of GPs.

#### 2.2.1. Resistance Among GPs

A significant proportion of South Tyrol’s GPs have expressed concerns about the transition to multidisciplinary care models. Many GPs value their autonomy and are wary of the potential loss of control over clinical decisions and patient management. The shift to collaborative teams may also increase their workload by requiring additional coordination, documentation, and communication efforts. These concerns are particularly acute among older GPs who have spent decades working independently and may be less inclined to adapt to new models of care [18].

#### 2.2.2. Balancing Continuity with New Care Models

The introduction of multidisciplinary teams and digital health tools, while intending to enhance care delivery, risks undermining the personal relationships that form the foundation of continuity in GP-led care. Team-based models often distribute responsibilities among various healthcare professionals, potentially diluting the direct patient–GP interaction that builds trust and familiarity. Similarly, the integration of digital tools, such as telemedicine and electronic health records (EHRs), may reduce face-to-face contact, further challenging the traditional dynamic of GP–patient relationships [12,13].

In a region characterized by an aging population, geographic disparities in healthcare access, and a bilingual and bicultural context [17], these challenges are compounded. Preserving continuity of care amidst these structural changes will require tailored strategies that respect the autonomy of GPs, leverage the potential of digital tools, and maintain the trusted role of GPs as the central point of contact in primary care [3].

## 3. Key Challenges

The Ministerial Decree 77/2022 reform seeks to modernize primary care by shifting GPs from independent practitioners to salaried employees working in multidisciplinary teams within CHCs [3]. However, staffing shortages and the slow operationalization of CHCs threaten the reform’s effectiveness and raise concerns about fragmented care. As of mid-2024, only 413 of the planned 1420 CHCs are operational. Many of these centers lack sufficient medical personnel, with some having no GPs or pediatricians. In addition, inconsistent physician availability in these facilities has raised concerns about their ability to function effectively. This slow progress stems from a combination of inadequate workforce planning, regional disparities in implementation, and limited alignment of resources with the reform’s ambitious goals [19].

### 3.1. Loss of Personal Connection

The transition to team-based care models risks disrupting long-standing patient–GP relationships. Traditionally, GPs act as trusted, singular contacts, often caring for multiple family generations, fostering trust, and a deep understanding of medical histories and personal contexts crucial for effective care [20]. Team-based models distribute responsibilities among various healthcare providers, potentially enhancing service range but weakening personal connections with GPs [21]. This shift may lead to a perceived loss of personalized attention, reducing patient trust and satisfaction [22].

When patients see multiple providers, there is a risk of inconsistent communication and fragmented care, which can undermine accountability and reliability in GP–patient relationships [23]. This may lead to poorer adherence to treatment plans and worse health outcomes. Addressing this issue is essential to maintaining the personal touch crucial to effective primary care.

### 3.2. Increased Bureaucracy for GPs

The use of digital platforms and multidisciplinary teams, intended to improve efficiency and care coordination, adds administrative burdens for GPs [24,25]. These duties include managing electronic health records (EHRs), coordinating with diverse team members, scheduling follow-ups, and documenting care plans. For GPs with high patient volumes, these tasks significantly reduce the time for direct patient care. Administrative duties can undermine GP–patient relationships, as GPs often struggle to balance documentation with meaningful interactions. This issue is significant in South Tyrol, where an aging GP workforce may struggle with new technologies and workflows. Insufficient training and support can result in frustration and burnout, and worsening workforce shortages.

The perception of increased bureaucracy might intensify GPs’ resistance to the proposed reforms, especially for those used to working independently. Streamlining administrative processes and providing adequate support staff for non-clinical tasks are essential to allow GPs to concentrate on patient care and uphold the trust crucial for primary care continuity.

### 3.3. Geographical and Cultural Barriers

South Tyrol’s unique geographical and cultural landscape [17] presents additional challenges to maintaining continuity of care during the transition to team-based models. The region’s rural and mountainous areas already face significant disparities in healthcare access, with limited availability of healthcare facilities and providers [26]. Ensuring that patients in these areas can consistently access their designated GPs or care teams is a logistical challenge exacerbated by the introduction of multidisciplinary models, which may require traveling to centralized CHCs.

The bilingual nature of South Tyrol adds another layer of complexity [27]. Effective communication is crucial for building trust and ensuring continuity of care, yet language barriers between patients and healthcare providers can obstruct relationship development. In team-based models, where patients interact with multiple providers, bilingual competence for the entire team is essential. Gaps in linguistic abilities risk alienating patients and disrupting their trust and comfort.

Cultural expectations of care, especially in multilingual areas like South Tyrol, exacerbate barriers to introducing new care models. Patients often prefer long-term, personal relationships with GPs who understand their sociocultural contexts. Failing to integrate cultural and linguistic factors when implementing new care models may lead to perceptions of depersonalized care, eroding trust and patient satisfaction, as demonstrated by misaligned role expectations between patients and GPs [28].

Reforms must prioritize equitable access to care in rural areas and support bilingualism across the healthcare workforce. Strategies to maintain continuity must be tailored to South Tyrol’s unique demographic, linguistic, and cultural context.

## 4. Proposed Strategies

To navigate the challenges of transitioning South Tyrol’s primary healthcare system to team-based care models, adapted strategies are required. These approaches must address the fragmented care delivery system, limited care management development, and hospital-centered primary care while fostering integration, collaboration, and gradual transformation [29,30,31].

To preserve continuity of care and trust in South Tyrol’s primary healthcare system during the transition to team-based care models, a set of strategic actions is essential. These strategies focus on maintaining strong GP–patient relationships, reducing administrative burdens, and fostering engagement with both patients and healthcare providers (Figure 2). The following section outlines these strategies, organized by overarching goals and their corresponding outcomes, to ensure an effective and patient-centered reform process. To enhance the methodological rigor of this study, the i-PARIHS framework is adopted to structure the proposed strategies [32]. I-PARIHS emphasizes that successful implementation depends on the interaction between the innovation (Ministerial Decree 77/2022 and its shift to team-based primary care), the recipients (GPs, healthcare professionals, and patients in South Tyrol), the context (South Tyrol’s bilingual, rural, and culturally distinct healthcare landscape), and facilitation (measures to support engagement and minimize disruption). By applying this framework, the recommendations are designed to enhance stakeholder readiness, maintain continuity of care, and optimize integration within the region’s healthcare system. This structured approach ensures that the transition to multidisciplinary primary care remains evidence-based, sustainable, and aligned with local needs.

### 4.1. Integration of Services

Establishing CHCs as hubs is crucial for integrating South Tyrol’s healthcare system, serving as central care points and coordinating rural clinics, and home-based programs. Care coordinators in CHCs facilitate collaboration among GPs, pediatricians, nurses, and hospital specialists, ensuring smooth transitions for patients with complex needs and enhancing care continuity. Nurse-led interventions play an important role in improving patient education, adherence, and chronic disease management. Studies show that nurses can effectively enhance self-management skills, optimize medication adherence, and provide continuity of care, particularly for elderly and multimorbid patients [33]. Their involvement in CHCs and home-based care reduces hospitalizations and supports preventive care, making them essential to the success of team-based primary care reforms.

Due to GPs’ resistance to fully integrate into team-based models, a gradual approach is essential. GPs should maintain autonomy while engaging with CHCs through virtual networks and collaborative tools. Optional participation in shared care planning and digital health initiatives allows GPs to experience integration benefits without coercion.

Nurses and allied health professionals should assume larger roles in care delivery [34]. Advanced training in chronic disease management enables nurses to serve as primary providers within CHCs for patients without GPs. Community health workers can address cultural and linguistic barriers, crucial in South Tyrol’s multilingual context, reducing GPs’ workload and improving healthcare accessibility and equity.

Effective communication across sectors is vital for integration. Shared EHRs improve information flow among GPs, CHCs, and hospitals. Multidisciplinary meetings, both virtual and in-person, enhance collaboration, and the primary–secondary care interface, fostering cohesive patient care.

To enhance integration, multidisciplinary teams should be supported by digital tools that facilitate collaboration while maintaining GP autonomy. Virtual teams can improve communication and decision-making without requiring physical presence. Shared care platforms provide real-time patient updates, and secure communication tools enable quick queries and coordination. Regular virtual meetings can discuss cases and align care, minimizing disruption to GPs’ routines. These strategies ensure teamwork complements traditional GP–patient relationships without compromising autonomy.

#### 4.1.1. Assign Primary Providers in Teams

Each patient should have a designated GP or primary provider within the multidisciplinary team to coordinate their care plan, monitor their health, and ensure continuity. A consistent provider maintains accountability and familiarity, ensuring personalized care and fostering trust.

In Italy, where GPs prefer independent office work, multidisciplinary teams can function as virtual teams [35,36]. These teams can use digital health platforms for communication, data sharing, and collaborative decision-making without needing to be physically co-located. For instance:Shared Care Platforms: Implement secure, interoperable EHRs that allow GPs to share patient information with other team members (e.g., specialists, nurses, and physiotherapists) in real time.Regular Virtual Meetings: Schedule periodic virtual case discussions via video conferencing to align care plans while minimizing disruption to GPs’ routines.Integrated Communication Tools: Use secure messaging systems for quick queries and updates among team members.Defined Roles and Responsibilities: Clearly delineate the role of each team member, ensuring GPs retain oversight while delegating specific tasks (e.g., health education, follow-ups) to other professionals.

Such a model combines the benefits of teamwork with the autonomy preferred by GPs, enabling comprehensive care without compromising the essence of traditional primary care relationships.

#### 4.1.2. Personalized Care Contracts

Personalized care contracts are essential for ensuring continuity and trust in primary care, particularly as health systems transition to multidisciplinary team models. These agreements define the roles and responsibilities of both the patient and their designated primary provider, who may be either a GP or a nurse.

In practice, GPs are likely to continue working predominantly in their own offices, maintaining long-term relationships with their existing patients. In contrast, nurses or GPs working full-time in CHCs are more likely to serve as primary providers for individuals without an established GP or those dissatisfied with their assigned provider. CHCs play a crucial role in addressing gaps in care, particularly in underserved communities, by leveraging the expertise of nurses and community health workers who facilitate access to services and support culturally competent care [37]. Studies underscore the importance of continuity and coordinated approaches in ensuring high-quality outcomes and patient satisfaction [38].

### 4.2. Building Infrastructure for Local Needs

Telemedicine and home-based care programs are crucial for South Tyrol’s healthcare infrastructure. Telemedicine effectively manages chronic diseases, post-treatment follow-ups, and preventive care, reducing the need for frequent in-person visits and enhancing accessibility for rural populations [39]. Home-based care programs by CHC teams address underserved patients’ needs by providing support at home.

A robust digital infrastructure is crucial for addressing South Tyrol’s healthcare challenges. User-friendly EHRs facilitate seamless patient information sharing among GPs, CHCs, and hospitals. Automation tools can alleviate administrative tasks like appointment reminders, data entry, and follow-ups, allowing healthcare providers to prioritize patient care. Developing standardized patient pathways ensures consistent care delivery, especially for chronic diseases. These pathways should be collaboratively designed with GPs to align with their clinical expertise and patient needs, promoting acceptance and enhancing care quality.

Additionally, AI-assisted decision support tools can enhance frailty risk assessment, automate clinical documentation, and improve diagnostic accuracy, reducing administrative burdens on GPs [40]. Beyond workflow efficiency, digital tools empower patients through medication reminders, remote monitoring, and personalized health education, improving engagement and adherence [41]. These advancements further streamline care coordination and enhance accessibility, ensuring that primary care reforms remain efficient and patient-centered.

### 4.3. Strengthening Engagement and Trust

Transparent communication with patients is crucial for managing expectations and building trust during this transition. Educational campaigns can outline the benefits of integrated care and emphasize patient-centered approaches in the reforms. Personalized discussions between patients and providers can address specific concerns, alleviating fears of losing trusted GP relationships. Feedback loops are essential for aligning care models with patient needs. Regular surveys and focus group insights can identify gaps and areas for improvement. Actively incorporating this feedback allows the healthcare system to refine workflows and communication strategies, ensuring ongoing progress and alignment with patient expectations.

### 4.4. Supporting General Practitioners

To ensure GPs remain engaged during the transition, their autonomy and clinical judgment must be preserved. Flexible participation models allow GPs to choose the extent of their collaboration with CHCs, enabling them to balance their independent practices with opportunities for shared care. Respecting their decision-making authority further reinforces their central role in the healthcare system.

Incentives are another critical factor in encouraging GP participation. Financial rewards, reduced administrative burdens, and opportunities for leadership within the new care structures can motivate GPs to actively engage in team-based care. By demonstrating tangible benefits, these incentives can help mitigate resistance and foster a collaborative environment.

Table 1 summarizes the proposed strategies and their intended outcomes. Together, these strategies provide a cohesive approach to addressing South Tyrol’s unique challenges, ensuring an effective and patient-centered transition to integrated primary care.

### 4.5. Benefits of Interprofessional Care Management

Team-based primary care improves patient safety, accessibility, and outcomes. Interprofessional collaboration reduces hospital readmissions, enhances medication adherence, and increases patient satisfaction [7,8,9]. Integrating nurses and allied health professionals strengthens chronic disease management, preventive care, and patient education, lowering costs and hospital dependency [6]. In South Tyrol, with its aging GP workforce and healthcare disparities, multidisciplinary teams are a key strategy to preserve continuity of care and enhance service delivery.

## 5. Monitoring and Evaluation

The full integration of healthcare reforms and team-based primary care models is a complex, multi-year process that requires coordinated efforts across policy, training, and practice levels [42,43,44]. While initial implementation can occur within a few years, achieving comprehensive integration and sustained improvements in healthcare delivery may take a decade or more, depending on the specific context and challenges faced.

Monitoring and evaluation are critical to ensuring the success of the proposed strategies and preserving continuity of care during the transition to team-based models. Tracking continuity metrics is essential to measure the rates of patient retention with their assigned GPs or care coordinators over time. Regularly monitoring the frequency of patient–GP interactions and the consistency of referral pathways within multidisciplinary teams will provide valuable insights into whether the new care models maintain continuity and where improvements may be needed [45].

Equally important is gauging the patient’s trust and satisfaction. Periodic surveys can evaluate trust in GPs, patient satisfaction with continuity, and the perceived ease of access to care. By including both quantitative measures and open-ended questions, these surveys will help identify patient concerns and ensure the reforms meet their expectations.

Understanding the experiences of GPs is vital for their sustained engagement. Feedback mechanisms such as surveys and interviews should assess workload changes, perceptions of autonomy, and overall job satisfaction. This input will help identify challenges related to administrative tasks, digital tool adoption, and team coordination, enabling timely adjustments to support GPs effectively.

The findings from these monitoring efforts should be used to refine the implementation process, creating a feedback loop that incorporates insights from both patients and GPs. This dynamic approach ensures that the reforms remain aligned with the goals of preserving continuity, fostering trust, and supporting the healthcare workforce. By systematically tracking and responding to these indicators, South Tyrol’s healthcare system can uphold its commitment to patient-centered care while successfully adapting to the mandates of Ministerial Decree 77/2022.

There is no empirical data on the impact of Ministerial Decree 77/2022 on primary care. Future research should assess patient and provider experiences, care continuity, and accessibility to guide policy adjustments and ensure a sustainable transition. Cost-effectiveness is a critical factor in the implementation of team-based primary care models. While initial investments in multidisciplinary teams, digital health tools, and CHCs may increase short-term costs, evidence suggests that these reforms can lead to long-term savings by reducing hospitalizations, improving chronic disease management, and enhancing preventive care [46,47]. Future assessments of Ministerial Decree 77/2022 could use Primary Care Assessment Tools (PCATools), a WHO-recommended framework based on Starfield’s principles [48]. PCATools measure accessibility, continuity, and coordination, providing insights to refine implementation and policy.

## 6. Conclusions

South Tyrol’s primary care reform under Ministerial Decree 77/2022 offers an opportunity to modernize healthcare delivery through multidisciplinary models, digital integration, and enhanced care coordination. However, the potential risks to GP–patient continuity and trust must be carefully managed to ensure that the strengths of the current system are not undermined.

By prioritizing strategies that preserve GP–patient continuity, the reform can maintain the strong personal relationships that underpin high-quality care in the region. Assigning primary providers within teams, minimizing administrative burdens, enhancing team dynamics, leveraging digital tools, addressing patient expectations, and supporting GPs through autonomy and incentives are all critical measures to achieve this balance.

With a robust monitoring framework to assess continuity, patient trust, and workforce engagement, the reform can remain dynamic and responsive to challenges. By aligning innovative care models with the principles of continuity and trust, South Tyrol can not only sustain but enhance its healthcare system, ensuring that the transition to multidisciplinary models strengthens the foundation of personalized, patient-centered care.

## Figures and Tables

**Figure 1 ijerph-22-00477-f001:**
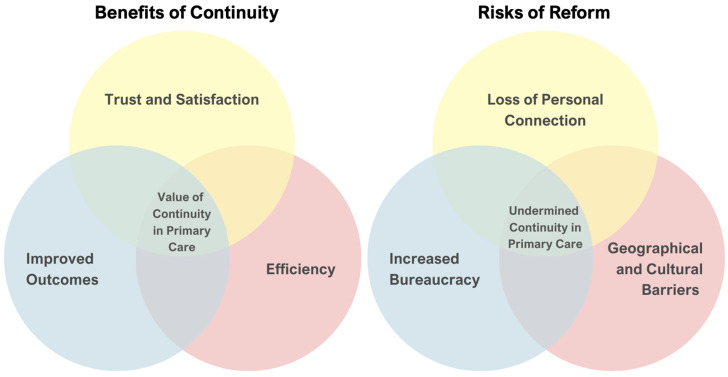
Continuity of care in South Tyrol: Balancing benefits and risks. The dual-layer Venn diagram illustrates the interplay between the key benefits of continuity in primary care and the risks posed by reforms under Ministerial Decree 77/2022 in South Tyrol. (**Left panel**) The benefits include trust and satisfaction derived from strong GP–patient relationships, improved health outcomes through better adherence to treatment, and enhanced efficiency by reducing healthcare costs and hospitalizations. The overlapping area highlights the value of continuity in primary care, where these benefits converge to provide comprehensive and patient-centered care. (**Right panel**) The risks to continuity include the loss of personal connection caused by fragmented care under team-based models, increased bureaucracy from digital tools and administrative tasks, and geographical and cultural barriers unique to South Tyrol’s rural and bilingual context. The intersection of these risks underscores the potential for undermined continuity in primary care, where multiple challenges disrupt effective care delivery.

**Figure 2 ijerph-22-00477-f002:**
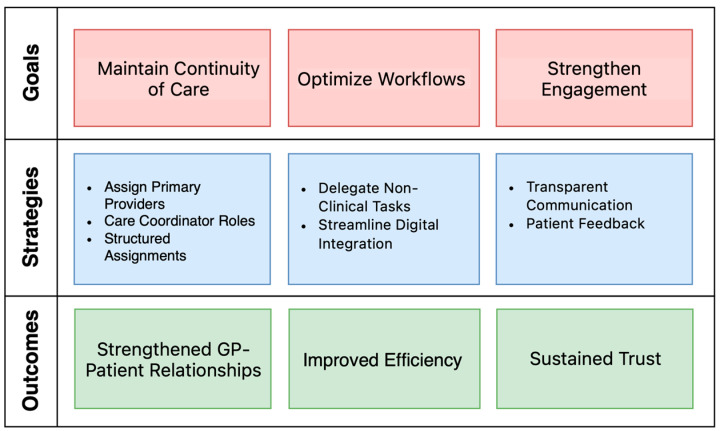
Framework of proposed strategies to preserve continuity in South Tyrol’s primary care reform.

**Table 1 ijerph-22-00477-t001:** Proposed strategies to maintain continuity and trust in South Tyrol’s primary care reform.

Goal	Strategy	Key Actions	Intended Outcomes
Maintain Continuity of Care	Assign Primary Providers in Teams	Designate a single GP or care provider as the main point of contact for each patient	Maintains trust, familiarity, and accountability in care
	Personalized Care Contracts	Develop agreements defining GP, nurse, and patient roles and responsibilities	Ensures clarity and confidence in maintaining personalized care
	Structured Patient Assignments	Create a system considering existing relationships, patient preferences, and geography	Reduces disruption and ensures equitable access to care
	Virtual Collaboration Tools	Use shared EHRs, secure messaging systems, and virtual meetings to support multidisciplinary teams	Enhances GP autonomy while enabling collaborative care
	Care Coordinator Roles	Appoint coordinators to manage complex cases, follow-ups, and multidisciplinary inputs	Frees GPs for clinical tasks while ensuring cohesive care delivery
Build Infrastructure for Local Needs	Telemedicine and Home-Based Care	Expand telemedicine services and home care programs to address rural and underserved populations	Improves accessibility and reduces dependence on in-person visits
	Standardized Patient Pathways	Collaborate with GPs to design pathways for chronic disease and preventive care management	Promotes consistent, high-quality care delivery
	Digital Integration	Implement user-friendly EHRs and automation tools to streamline administrative tasks	Enhances workflow efficiency and allows providers to focus on care
Strengthen Engagement and Trust	Transparent Communication	Conduct education campaigns and personalized discussions to explain reforms	Builds trust and reduces anxiety about the transition
	Incorporate Patient Feedback	Use surveys and focus groups to gather insights and improve care models	Aligns reforms with patient needs and fosters engagement
Support General Practitioners	Preserve Autonomy	Allow GPs to choose participation levels, maintain clinical judgment, and engage flexibly with CHCs	Reduces resistance to reforms and ensures satisfaction
	Incentives for Participation	Provide financial rewards, career development, and public recognition for GPs maintaining continuity	Encourages active participation and reinforces trust in reforms

Abbreviations: CHCs, Community Health Centers; EHRs, electronic health records; GPs, general practitioners.

## Data Availability

No new data were created.

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
