# Peer review of "Preserving Continuity and Trust in Primary Care: Strategies for Implementing Team-Based Models in South Tyrol, Italy"

_ijerph, 2025, doi:10.3390/ijerph22040477_

Round 1
Reviewer 1 Report
Comments and Suggestions for Authors
Thank you for the opportunity to peer review this important work. The followings are recommendations to make the manuscript more approachable and appreciative of the insights offered by an international readership.
- The author has made several references to the Ministerial Declaration. There are bits and pieces on the impacts of the Declaration scattered across the manuscript. A more thorough and coherence introduction to WHAT the Declaration is and its impacts on general practice, as well as the uniqueness of South Tyrrol would be very beneficial to international readers. Without such CONTEXT, the generalisability/ transferability of this work may be unintentionally limited.
- The author may have adopted an implementation framework such as i-PARIHS or CFIR frameworks in framing their recommendation. The HOW (the process) would greatly increase the validity/ research rigour of this work.
Reviewer 2 Report
Comments and Suggestions for Authors
The publication deals with the implementation of new solutions, team-based models in healthcare. The psychological, social barriers to implementation in relation to the regional situation are described. the importance of changing the benefits of, among other things, the role of the interprofessional team and the importance of the team in terms of safety and continuity of care with an ageing physician population is not sufficiently documented. Evidence that interprofessional team-based care definitely increases accessibility, safety and health management by patients. ICT systems reduce bureaucracy as well as issues related to the cultural theme. The introduction of, for example, nurses into patient care also provides a better guarantee for patients to understand and manage their health problems better. System changes will be evident, according to the meta-analysis in maybe an average of 17 years, by which time there will be a different population in the system, more oriented towards other services not only related to direct contact. The cost-effectiveness of implementing the solutions is also not indicated, It seems that the publication would benefit from a balance between costs and benefits.
Reviewer 3 Report
Comments and Suggestions for Authors
The subject is relevant.
Primary health care is the main tool for improving people's quality of life, especially with the advancement of ageing. It also guarantees savings for the health system.
The introduction and background could be reformulated to provide a characterization of what health care is like in Tyrol (from what is understood in the text, there is a monopoly of general practitioners) and what the Ministerial Decree will change, in addition to the accessibility barriers encountered (described in the article in section 3 Main Challenges). The way it is reported leads the reader to understand that the change will be bad for the population, contrary to the literature and the experience of several countries such as Canada and Brazil.
The proposed strategies are interesting and are in line with the principles of primary health care.
There has been no study on the implementation of the 2022 Decree. Ideally, an operational study should have been carried out to assess client and professional satisfaction. There are still tools such as the Primary Care Assessment Tool (PCATools) based on the Starfield principles, recommended by the WHO, to assess the attributes of Primary Health Care.
Author Response
Comment 1: Clearly define South Tyrol’s existing GP-centered system and how MD 77/2022 changes
Response 1: Reviewer 1 had a similar request. Following text and references were inserted in the Intrioduction section in response replacing in part other text (:
“However, the introduction of Ministerial Decree 77/2022 represents a significant trans-formation in the country’s primary care landscape. This reform mandates the estab-lishment of multidisciplinary Community Health Centers (Case di Comunità, CHCs) and promotes a shift toward team-based care models, aiming to address workforce shortages, enhance healthcare accessibility, and integrate digital health solutions into routine primary care delivery [3]. Ministerial Decree 77/2022, issued on June 22, 2022, reforms Italy's National Health Service (NHS) by shifting from hospital-centered care to decen-tralized, community-based healthcare [4]. A key aspect is the establishment of Com-munity Health Centers (Case di Comunità, CHCs), designed to operate 24/7 and integrate primary, specialist, and social care. The decree mandates over 1,350 CHCs nationwide, staffed by multidisciplinary teams, including general practitioners, specialists, nurses, and social workers, to enhance accessibility and continuity of care. The reform also promotes digital health solutions, such as electronic health records and telemedicine, to improve care coordination, especially in underserved areas. By strengthening primary care, preventive medicine, and team-based models, the decree aims to reduce hospi-talizations and foster a more resilient, patient-centered healthcare system in Italy.
While Ministerial Decree 77/2022 is a national reform, its implementation in South Tyrol—a uniquely autonomous, bilingual region in northern Italy —presents distinct challenges and opportunities. South Tyrol’s healthcare system operates within a cultural and linguistic context that differs from much of Italy, with a significant proportion of the population being German-speaking and a tradition of strong, long-term relationships between GPs and their patients [3]. The shift from independent GP-led care to CHC-based multidisciplinary teams risks disrupting these well-established relationships, introducing potential barriers related to language, bureaucracy, and patient trust [5].”
Comment 2: Acknowledge both challenges and benefits, referencing Canada and Brazil as examples.
Response 2: Following text and references were inserted in the Intrioduction section in response:
“International experience shows that team-based primary care can improve access, continuity, and efficiency [8]. In Canada, Primary Care Networks and Family Health Teams have enhanced chronic disease management, reduced hospital visits, and strengthened preventive care [10]. Brazil’s Family Health Strategy has expanded primary care access and improved population health outcomes [11]. While Ministerial Decree 77/2022 presents structural challenges, its success in South Tyrol will depend on effective team integration, digital tools, and patient-centered strategies, balancing risks with proven benefits.“
Comment 3: Highlight the absence of empirical studies on the decree’s impact.
Response 3: Following text was added to the end of the “Monitoring and Evaluation” section before the “Conclusions”:
“There is no empirical data on the impact of Ministerial Decree 77/2022 on primary care. Future research should assess patient and provider experiences, care continuity, and accessibility to guide policy adjustments and ensure a sustainable transition.”
Comment 4: Mention PCATools as a useful framework for future assessment.
Response 4: Following text and reference was added to the “Monitoring and Evaluation” section before the “Conclusions”:
“Future assessments of Ministerial Decree 77/2022 could use Primary Care Assessment Tools (PCATools), a WHO-recommended framework based on Starfield’s principles [48]. PCATools measure accessibility, continuity, and coordination, providing insights to refine implementation and policy.”
Round 2
Reviewer 1 Report
Comments and Suggestions for Authors
Thank you for considering the comments. Look forward to seeing this work published and follow on the impact of your work especially how it may further inform the rollout of the Declaration.
Reviewer 3 Report
Comments and Suggestions for Authors
Dear authors,
The topic is relevant. The process of implementing decentralization and multiprofessional and interdisciplinary care in primary healthcare is an important challenge for public policies. The quality and clarity of the manuscript have improved, and the previously suggested recommendations have been adopted.
Thank you for the corrections.